# MTEEG: A Multi-Task Learning Framework for Enhanced Electroencephalography Analysis Using Low-Rank Adaptation

## Abstract

Electroencephalography (EEG) analysis using deep learning has traditionally placed a strong emphasis on models that are custom-built and optimized for specific datasets. Several recent research utilize self-supervised learning to extract generic representations from massive amounts of unlabeled EEG data. The pre-trained models are then fine-tuned on each downstream dataset independently, demonstrating promising results. However, in practical applications involving multiple tasks, utilizing a separate model for each is not ideal regarding computational and spatial cost. In this study, we go one step further and explore the simultaneous adaptation of a pre-trained model to multiple different tasks. The EEG signals exhibit significant heterogeneity due to their collection from various subjects using diverse devices and experimental setups, resulting in potential conflicts among different tasks that impede joint optimization. To tackle this challenge, we propose MTEEG, a multi-task EEG recognition framework which incorporates a task-agnostic temporal encoder and task-specific low-rank adaptation modules to disentangle the parameter space, facilitating both task interaction and specification. Experiments show that MTEEG surpasses other multi-task methods and performs on par with state-of-the-art single-task methods on abnormal detection, event type classification, emotion recognition, seizure detection, sleep stage classification and motor imagery classification after being tuned jointly on six publicly available datasets. MTEEG shows the potential of multi-task EEG recognition and promotes the development of general-purpose brain-computer interfaces in the future. The source code will be released.

## 1 Introduction

Electroencephalography (EEG) is a widely used neuroimaging technique that captures electrical activity of the brain through non-invasive scalp electrodes. In recent years, deep learning models, such as convolutional neural networks (CNNs) and transformers, have demonstrated remarkable success in extracting meaningful patterns from EEG data, leading to significant improvements in various applications including emotion recognition (Li et al., 2022b), motor imagery classification (Li et al., 2022b) and seizure detection (Boonyakitanont et al., 2020). However, despite their power, these models are typically customized for specific tasks and input formats, which causes them to overfit and become ungeneralizable.

Drawing inspirations from the advancements of large language models (Devlin, 2018; Achiam et al., 2023), some researchers (Yang et al., 2023a; Yi et al., 2024; Jiang et al., 2024) employ self-supervised learning to extract generic representations from large amounts of unlabeled EEG data, significantly improving the model's generalizability. Despite their remarkable performance, these models necessitate individual fine-tuning for each downstream dataset, thereby constraining their versatility and applicability in practical scenarios involving multiple tasks. For example, an EEG-based health monitoring system may need to perform and switch between seizure detection, emotion recognition and sleep stage classification per demand to have a comprehensive evaluation of the patient's condition, both physically and mentally. In this case, a pre-trained model must be replicated and fine-tuned three times, once for each task, resulting in significant computational and spatial

overhead. Therefore, it would be beneficial to have a unified system that is capable of handling different tasks simultaneously.

Despite the promise, challenges persist to build an efficient multi-task model for EEG processing. The EEG signals, collected from various subjects utilizing different devices and experimental configurations, exhibit markedly distinct intrinsic characteristics. This variability can mislead the model with conflicting parameter update directions, leading to a substantial decrease in learning efficacy. Similar heterogeneity-induced issues have also been noted in other domains (Yu et al., 2020; Zhou et al., 2024b), and many methods have been proposed to tackle them; some incorporate separate modules for specific tasks (Liu et al., 2022b; Mahabadi et al., 2021), while others use soft-gating mechanisms to flexibly assign modules for different tasks (Ma et al., 2018; Cheng et al., 2016). Nevertheless, the majority of these studies focus on the analysis of image, text and audio data, raising doubts about the applicability of their findings to EEG.

In this study, we propose MTEEG, a novel EEG recognition framework which exploits a pre-trained LaBraM (Jiang et al., 2024) along with task-specific modules to facilitate efficient multi-task joint training. It consists of three major components: 1) a temporal encoder that's shared across all the tasks; 2) a transformer encoder with a frozen shared backbone and multiple task-specific low-rank adapters; 3) task-specific classification heads that output the final predictions. During training, the task-agnostic temporal encoder promotes interaction among different tasks and the reuse of global knowledge, whereas the transformer encoder allocates specialized low-rank adapters to each task, explicitly isolating the parameters. Thus, the disentanglement of task-specific knowledge towards their corresponding adapters effectively reduces conflicts arising from heterogeneity. Furthermore, since the task-specific modules are implemented with low-rank adapters, the computational and spatial overhead they incur is significantly lower than that of fully fine-tuning a pre-trained model. In summary, our contributions are as follows:

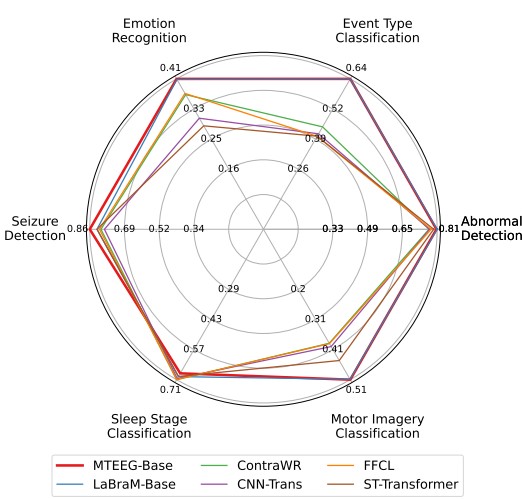

Figure 1: Overview of MTEEG's performance (balanced accuracy) on downstream datasets.

- We investigate multi-task EEG recognition, which is a crucial yet underexplored aspect in the practical application of brain-computer interfaces. Concurring with prior research on other data types, we observe that joint training on heterogeneous EEG datasets also presents the issue of conflicts between different tasks, leading to substantial performance deterioration of the model.

- We present the MTEEG framework, which enhances a pre-trained model by incorporating task-specific modules to achieve parameter isolation across different tasks. This isolation allows for the separation of gradients to prevent conflicts, hence facilitating efficient multi-task joint training.

- Through extensive experiments, we demonstrate that after joint optimization on six publicly available datasets, MTEEG can handle abnormal detection, event type classification, emotion recognition, seizure detection, sleep stage classification and motor imagery simultaneously, achieving performance superior than other multi-task methods and on par with state-of-the-art single-task methods.

## 2 RELATED WORK

**Self-supervised EEG pre-training**. Despite the scarcity of annotated EEG data, there is a substantial volume of unlabeled EEG data collected from various sources. Consequently, there has been a growing interest in adopting self-supervised methods to learn generic representations from these

unlabeled data to improve the model's performance and generalizability. BENDR (Kostas et al., 2021) utilizes a contrastive learning model, wav2vec 2.0 (Baevski et al., 2020), to learn compressed representations of raw EEG signals. Neuro-GPT (Cui et al., 2024) masks random parts of the input and lets the model learn to reproduce the original signal. Brant-2 incorporates both mask-prediction and forecasting pretext tasks to enhance the model's robustness and scalability. EEG2Rep (Mohammadi Foumani et al., 2024) reconstructs the masked samples in an abstract representation space to enhance the semantic quality of EEG representations. MMM (Yi et al., 2024) spatially divides the scalp into 17 regions and allocate a learnable token to each of them, enabling a unified topology for cross-dataset pre-training. LaBraM (Jiang et al., 2024) learns common spatial embeddings based on the 10-20 international system to be compatible with different electrode configurations, and adopts a two-stage pre-training paradigm to facilitate representation learning from noisy EEG signals.

**Multi-task learning**. Multi-task learning (MTL) aims to develop a model capable of handling various tasks simultaneously. The existing methods for MTL differ in how and where different tasks interact with each other. Hard parameter sharing (HPS) methods (Long et al., 2017; Lu et al., 2017) employ a single encoder for all tasks, resulting in exceptional scalability but limitations in their ability to deal with the conflicts between different tasks. The cross-stitch network (Misra et al., 2016) introduces a sharing unit to linearly combine the activation values at each layer. MTAN (Liu et al., 2019) uses attention modules to compute attention masks, thereby controlling the parameters involved in processing each task. MMoE (Ma et al., 2018) proposes to share multiple experts among different tasks with weights computed by task-specific gates, thus enabling the model to automatically learn how to balance the experts given specific inputs. PLE (Tang et al., 2020) explicitly divides experts into shared and task-specific ones, further improving the model's robustness. In addition to the aforementioned methods that specifically target image processing, the concept of MTL has also been incorporated into EEG analysis. MIN2Net (Autthasan et al., 2021) and ERPENet (Ditthapron et al., 2019) utilize multi-task autoencoder to achieve good performance on motor imagery and P300 classification, respectively. GMSS (Li et al., 2022c) constructs different pretext tasks for a graph-based self-supervised learning model to reduce the chance of overfitting. These methods are fundamentally different from MTEEG in that they hand-craft tasks to serve for better optimization on a single dataset, while MTEEG is designed to be jointly optimized on heterogeneous datasets.

**Low-rank adaptation**. Low-Rank Adaptation (LoRA) (Hu et al., 2021) is a parameter-efficient fine-tuning method, which aims at reducing space and computation cost without sacrificing the model's expressiveness. It has been widely used for adapting large foundation models to specific domains (Zhang et al., 2023; Zhou et al., 2024a). In the context of MTL, LoRA has also shown great potential because of its high level of flexibility. LoraHub (Huang et al., 2023) combines multiple LoRA modules to enhance cross-task generalization in few-shot scenarios. MOELoRA (Liu et al., 2023) integrates LoRA into a Mixture-of-Experts (MOE) framework and demonstrates superior performance. LoRAMOE (Dou et al., 2024) utilizes LoRA as an MOE-style plugin to alleviate the world knowledge forgetting problem in large language models. MoLA (Zhou et al., 2024b) includes LoRA during the training procedure and verifies their method on multiple types of heterogeneous data. However, unlike MTEEG which targets a cross-dataset setting, these methods are still limited to tasks within the same dataset.

## 3 METHOD

### 3.1 PROBLEM FORMULATION

Assume there are a total of $P$ datasets. For $p \in \{1, 2, \ldots, P\}$, given any multi-channel EEG signal $X \in \mathbb{R}^{C_p \times T_p}$ in the $p$-th dataset, where $C_p$ and $T_p$ represent the number of channels and the input duration respectively, the model aims to predict the corresponding label $y \in \mathcal{Y}_p$, where $\mathcal{Y}_p$ represents the set of all possible outputs.

### 3.2 MODEL ARCHITECTURE

The architecture of MTEEG is built upon that of LaBraM. An input EEG sample $X \in \mathbb{R}^{C_p \times T_p}$ is first segmented in the temporal dimension with a non-overlapping window of length $w$, resulting in patches $\boldsymbol{x} = \{x_{i,j} | i = 1, 2, \ldots, C_p, j = 1, 2, \ldots, \lfloor \frac{T_p}{w} \rfloor\}$. The patches are then processed se-

quentially by the temporal encoder, transformer encoder and classification head to produce the final output.

**Temporal Encoder**. The temporal encoder takes the segmented input patches and encode them into embeddings, serving to capture the intricate temporal features in the signal. It consists of multiple temporal convolution blocks, each of which is composed of a 1-D convolution layer, a group normalization layer, and a GELU activation function. Formally, given a set of input patches $x$ from dataset $p$, the output can be denoted as

$$\{e_{i,j} = TE(x_{i,j}) \in \mathbb{R}^d | x_{i,j} \in x, i = 1, 2, \ldots, C_k, j = 1, 2, \ldots, \lfloor \frac{T_p}{w} \rfloor\},$$

where $TE$ represents the temporal encoder and $d$ is the dimension of the embeddings.

**Transformer Encoder**. To take account of the global features in the signal, we add the patch embeddings with temporal and spatial embeddings based on the 10-20 international system, then feed them into the transformer encoder to be processed with the attention mechanism. The attention function can be formulated as

$$\text{Attention}(Q, K, V) = \text{softmax}(\frac{\text{LN}(Q)\text{LN}(K)^T}{\sqrt{d_p}})V,$$

where $d_p$ is the dimension of the key and query, and LN stands for layer normalization, which are added to stabilize training by avoiding overly large values in the attention logits.

Following common practice, we employ multi-head attention to let the model attend to information from different representational subspaces:

$$\text{MultiHead}(Q, K, V) = \text{Concat}(\text{head}_1, \ldots, \text{head}_h)W^O$$
$$\text{where head}_i = \text{Attention}(QW_i^Q, KW_i^K, VW_i^V)$$

where $h$ is the number of heads, $W_i^Q \in \mathbb{R}^{d_{\text{model}} \times d_k}$, $W_i^K \in \mathbb{R}^{d_{\text{model}} \times d_k}$, $W_i^V \in \mathbb{R}^{d_{\text{model}} \times d_v}$, $W_O \in \mathbb{R}^{hd_v \times d_{\text{model}}}$ are the linear projection matrices.

### 3.3 TRAINING PROCEDURE

The training of MTEEG entails a two-stage process. In the first stage, a LaBraM model is pre-trained on unlabeled data to provide a solid foundation for extracting useful information raw EEG signals. Specifically, we start by training a neural tokenizer which is inspired by VQ-VAE (Van Den Oord et al., 2017). The tokenizer employs the architecture outlined in Section 3.2 and is followed by a neural codebook which quantizes the continuous representations into discrete tokens. The learning process is then guided by the reconstruction of the amplitude and phase from these discrete tokens. After the tokenizer is sufficiently trained, we train the LaBraM model by randomly masking a proportion of the input patches and letting the model predict their corresponding indices in the codebook. Some technical details are omitted here since the pre-training stage is not the main focus of this work.

In the second stage, the pre-trained model is adapted to downstream datasets via a fine-tuning process, in which we incorporate two major designs. Firstly, the parameters of the temporal encoder are shared across and updated by all the tasks to promote the reuse of global knowledge. Secondly, in the transformer encoder, we allocate specialized low-rank adapters to each task to achieve parameter isolation. An overview of the fine-tuning stage is shown in Figure 2. For any linear layer $f$ with weight matrix $W_0 \in \mathbb{R}^{m \times n}$ and bias $b_0$, we define a set of low-rank decomposition matrices $\Delta W = \{\Delta W_p = B_p A_p | B_p \in \mathbb{R}^{m \times r}, A_p \in \mathbb{R}^{r \times n}, p = 1, 2, \ldots, P\}$ where $r$ is the rank and $P$ is the total number of tasks. When the model performs the $p$-th task, the corresponding adapter is injected into the layer and the original linear operation is transformed into

$$f(x) = W_0 x + \Delta W_p x + b_0$$
$$= (W_0 + B_p A_p)x + b_0$$

We apply this transformation to the linear projections of query, key, value and output matrices, as well the fully connected feed-forward network that follows the attention layers. Formally, for task $p$, the output of a single attention head is

$$\text{head}_i = \text{Attention}(Q(W_i^Q + B_{i,p}^Q A_{i,p}^Q), K(W_i^K + B_{i,p}^K A_{i,p}^K), V(W_i^V + B_{i,p}^V A_{i,p}^V))$$

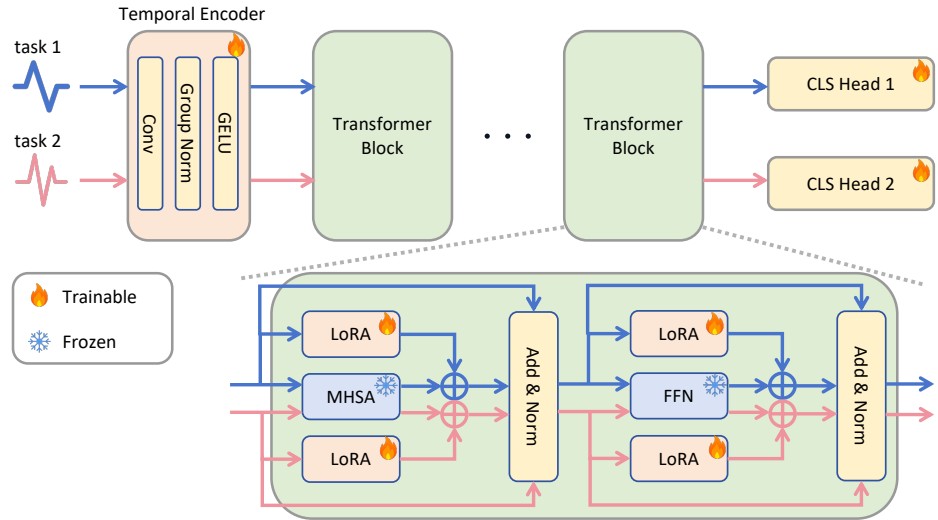

Figure 2: Overview of the fine-tuning stage. The temporal encoder, task-specific low-rank adapters and classification heads are trainable, while the pre-trained weights in the transformer encoder remain frozen.

and the full multi-head attention can be rewritten as

$$\text{MultiHead}(Q, K, V) = \text{Concat}(\text{head}_1, \ldots, \text{head}_\text{h})(W^O + B_p^O A_p^O)$$

where $h$ is the number of heads, $W_i^Q$, $W_i^K$, $W_i^V$, $W_O$ are the pre-trained weights for linear projections and $B_{i,p}^Q A_{i,p}^Q$, $B_{i,p}^K A_{i,p}^K$, $B_{i,p}^V A_{i,p}^V$, $B_p^O A_p^O$ are the corresponding task-specific low-rank adapters.

Throughout the fine-tuning stage, all the pre-trained weights in the transformer encoder are kept frozen and only the low-rank adapters are trainable. In this way, the gradients from different tasks are distinctly separated and confined within different modules, thereby alleviating the heterogeneous conflict issue.

## 4 EXPERIMENTS

### 4.1 DOWNSTREAM DATASETS

After pre-training, we fine-tune and evaluate our MTEEG jointly on the following six datasets, the statistics of which are detailed in Table 1.

**TUAB** (abnormal detection) (Obeid & Picone, 2016): A corpus of EEGs that have been annotated as normal or abnormal.

**TUEV** (event type classification) (Obeid & Picone, 2016): A subset of TUEG that contains annotations of EEG segments as one of six classes: (1) spike and sharp wave (SPSW), (2) generalized periodic epileptiform discharges (GPED), (3) periodic lateralized epileptiform discharges (PLED), (4) eye movement (EYEM), (5) artifact (ARTF) and (6) background (BCKG).

**SEED-V** (emotion recognition) (Liu et al., 2021): An emotion EEG dataset collected while 16 subjects watched video clips corresponding to five emotion categories (happy, sad, neutral, disgust, and fear).

**CHB-MIT** (seizure detection) (Shoeb, 2009): A database from Children's Hospital Boston consisting of EEG recordings from 22 pediatric subjects with intractable seizures. Signals are sampled with 23 bipolar channels and we select the 16 standard montages in the experiments. Since the dataset is highly imbalanced (about 0.3% positive ratio), we segment the seizure regions with a 1-second stride to generate overlapping samples. In addition, we follow common practices (Lee et al., 2024; Chung et al., 2024) to randomly select 10% of the negative samples during training.

**Sleep-EDF** (sleep stage classification) (Goldberger et al., 2000): A database containing 197 whole-night PolySomnoGraphic sleep recordings, among which we use the 153 recordings from the study of age effects in healthy subjects (SC) in the experiments. Samples are manually annotated as one of the eight classes (W, N1, N2, N3, N4, REM, MOVEMENT, UNKNOWN). Following previous works (Supratak et al., 2017; Supratak & Guo, 2020), we exclude movement artifacts at the beginning and the end of each sleep data that was labeled as MOVEMENT or UNKNOWN, as they do not belong to the five sleep stages. In addition, we merge the N3 and N4 stages into a single stage N3 to stick to the AASM manual (Berry, 2012).

**PhysioNet** (motor imagery classification) (Goldberger et al., 2000): A dataset containing EEG recordings from 109 participants, with trials that belong to 5 classes: left hand, right hand, both hands, both feet, as well as rest. Following previous works (Barmpas et al., 2023; Zoumpourlis & Patras, 2024), we discard data from 6 participants (S088, S090, S092, S100, S104, S106) that have inconsistent sampling frequencies or trial lengths.

Table 1: Downstream dataset statistics

| Dataset | # Channel | Sampling Rate (Hz) | Duration (seconds) | # Sample | Task |
| --- | --- | --- | --- | --- | --- |
| TUAB | 23 | 256 | 10 | 409,455 | Binary classification |
| TUEV | 23 | 256 | 5 | 112,491 | 6-class classification |
| SEED-V | 62 | 1000 | 1 | 148,694 | 5-class classification |
| CHB-MIT | 16 | 256 | 10 | 26,483 | Binary classification |
| Sleep-EDF | 2 | 100 | 30 | 195,479 | 6-class classification |
| PhysioNet | 64 | 160 | 4 | 18,540 | 5-class classification |

## 4.2 EXPERIMENTAL SETUP

**Preprocessing**. We first filter the EEG signals within the range of 0.1 Hz to 75 Hz to eliminate low-frequency noise. A 50 Hz notch filter is subsequently employed to eliminate power-line interference. After that, all EEG signals are resampled to a frequency of 200 Hz. The typical range of EEG values is between -0.1 mV and 0.1 mV, which we normalize by setting the unit to 0.1 mV to ensure the values predominantly fall between -1 and 1.

**Pre-training & Fine-tuning**. We construct MTEEG utilizing two different configurations of LaBraM, specifically LaBraM-Base and LaBraM-Large, yielding MTEEG-Base and MTEEG-Large correspondingly. For the pre-training of LaBraM, We use the default hyperparameters outlined in the original paper. The pre-training data comprises nine public datasets, detailed in Appendix A, with a total duration of approximately 2000 hours. In the fine-tuning stage, the datasets are first split into training, validation and test subsets as outlined in Appendix B. Subsequently, we train the models using binary cross-entropy loss for binary classification tasks and cross-entropy loss for multi-class classification tasks. Due to the significantly larger data volume of TUAB compared to other datasets, which leads to early convergence and overfitting, we randomly sample 10% of the data points in TUAB for each training epoch to balance the optimization. All the experiments are conducted on Linux servers equipped with NVIDIA A100 GPUs and Python 3.10.14 + PyTorch 2.2.2 + CUDA 12.1 environment. The optimal models are trained on the training set, selected from the validation set, and finally evaluated on the test set. We report the average and standard deviation values on three different random seeds to obtain comparable results.

**Baselines**. For single-task baselines, we consider both self-supervised and supervised methods. Self-supervised baselines include LaBraM and BIOT (Yang et al., 2023a). Supervised baselines include SPaRCNet (Jing et al., 2023), ContraWR (Yang et al., 2021), CNN-Transformer (Peh et al., 2022), FFCL (Li et al., 2022a) and ST-Transformer (Song et al., 2021). LaBraM and BIOT are publicly accessible in their official repositories, with the supervised methods implemented by BIOT. We use the default hyperparameters for fair comparison.

Given that multi-task learning in EEG processing is underexplored and there is currently no public method for comparison, we integrate a pre-trained LaBraM-Base as the backbone network within three established multi-task learning frameworks to set up the multi-task baselines. These frame-

works include: (1) HPS (Long et al., 2017; Lu et al., 2017) where different tasks share the same expert (backbone network), except for the classification heads, (2) MMoE (Ma et al., 2018) where multiple experts are shared among different tasks with weights controlled by task-specific gates, (3) CGC (Cheng et al., 2016) where both shared and task-specific experts are included to enhance the extraction of heterogeneous features. The implementation is based on LibMTL (Lin & Zhang, 2022). Following common practice, we set the number of shared experts in MMoE and CGC to match the number of tasks, which is six in our case, and we designate one task-specific expert per task in CGC.

**Metrics**. We use the following metrics for evaluating the models: (1) Balanced Accuracy: the average of recall (sensitivity) on each class. (2) AUC-PR: area under the precision-recall curve, which summarizes the trade-off between precision and recall at different classification thresholds. This metric is used for binary classification. (3) AUROC: area under the receiver operating characteristic curve, which summarizes the trade-off between the true positive rate (sensitivity) and the false positive rate (1-specificity) at different classification thresholds. This metric is used for binary classification. (4) Cohen's Kappa: an assessment of the agreement between two classifiers on a categorical scale, taking into account the possibility of agreement occurring by chance. This metric is used for multi-class classification. (5) Weighted F1: a weighted average of individual F1-scores for each class. This metric is used for multi-class classification. AUROC and Cohen's Kappa are used as the monitoring metrics for binary and multi-class classifications respectively. For multi-task methods, we monitor the average values of these metrics across all tasks. We use PyHealth (Yang et al., 2023b) for the implementation of all the metrics.

### 4.3 COMPARISON WITH OTHER METHODS

The main results are summarized in Table 2, 3 and 4. The best results of multi-task and single-task methods in each column are highlighted in bold and underlined, respectively. Based on these results, we make the following observations.

Firstly, there exists a significant performance gap between HPS and LaBraM-Base across all tasks and metrics, despite their architectural similarities. This suggests that, similar to other data types, EEG signals from diverse sources can also confuse the model due to conflicting optimization directions, resulting in substantial performance degradation. Although multi-task methods such as MMoE and CGC have demonstrated efficacy in addressing this issue in other domains, their effectiveness in EEG processing remains limited. This may result from the gating mechanism in these methods being implemented with basic linear layers, which may be inadequate for differentiating the intricate intrinsic properties of highly noisy EEG signals. Secondly, in comparison to its multi-task counterparts, our proposed MTEEG-Base exhibits comparable performance on SEED-V and significantly outperforms them across all other datasets, thereby demonstrating the efficacy of gradient separation with task-specific low-rank adapters. Moreover, MTEEG even performs on par with the state-of-the-art single-task method. Comparing to LaBraM-Base, MTEEG-Base performs better on TUEV, SEED-V, CHB-MIT, and PhysioNet and slightly worse on TUAB and Sleep-EDF. The same phenomenon is also evident in the large variant of the model, confirming the scalability of our approach. Thirdly, MTEEG has the advantage of being lightweight. The base and large variants have only 1.8M and 7.4M trainable parameters fine-tuning respectively, compared to 5.8M and 46M for LaBraM-Base and LaBraM-Large. The time and space efficiency associated with this lightweight design would be beneficial in practical applications, particularly when computational resources are constrained or latency is critical.

### 4.4 ABLATION STUDIES

Ablation studies were performed on all six datasets; however, results are only presented for TUAB, TUEV, and SEED-V in the main paper to conserve space. For additional results on the other datasets, please refer to Appendix C.

**Impact of adapter rank** $r$. We assign different values to $r$, ranging from 4 to 32 to examine its impact on the model's downstream performance. As illustrated in Figure 3, the base variant consistently achieves its maximum performance at $r = 8$ across all datasets, whereas the large variant reaches peak performance at $r = 16$ on TUAB and $r = 8$ on the remaining datasets. This indicates that a higher rank does not necessarily yield better performance, likely due to over-fitting

Table 2: Results on TUAB and TUEV

| Methods | # Trainable Parameters | TUAB | | | TUEV | | |
|---|---|---|---|---|---|---|---|
| | | Balanced Acc. ↑ | AUC-PR ↑ | AUROC ↑ | Balanced Acc. ↑ | Cohen's Kappa ↑ | Weighted F1 ↑ |
| Single-task methods | | | | | | | |
| SPaRCNet | 0.79M | 0.7896±0.0018 | 0.8414±0.0018 | 0.8676±0.0012 | 0.4161±0.0262 | 0.4233±0.0181 | 0.7024±0.0104 |
| ContraWR | 1.6M | 0.7746±0.0041 | 0.8421±0.0104 | 0.8456±0.0074 | 0.4384±0.0349 | 0.3912±0.0237 | 0.6893±0.0136 |
| CNN-Transformer | 3.2M | 0.7777±0.0022 | 0.8433±0.0039 | 0.8461±0.0013 | 0.4087±0.0161 | 0.3815±0.0134 | 0.6854±0.0293 |
| FFCL | 2.4M | 0.7848±0.0038 | 0.8448±0.0065 | 0.8569±0.0051 | 0.3979±0.0104 | 0.3732±0.0188 | 0.6783±0.0120 |
| ST-Transformer | 3.5M | 0.7966±0.0023 | 0.8521±0.0026 | 0.8707±0.0019 | 0.3984±0.0228 | 0.3765±0.0306 | 0.6823±0.0190 |
| BIOT | 3.2M | 0.7959±0.0057 | 0.8792±0.0023 | 0.8815±0.0043 | 0.5281±0.0225 | 0.5273±0.0249 | 0.7492±0.0082 |
| LaBraM-Base | 5.8M | 0.8126±0.0019 | 0.8911±0.0090 | 0.8843±0.0102 | 0.6436±0.0031 | 0.6254±0.0157 | 0.8172±0.0063 |
| LaBraM-Large | 46M | 0.8137±0.0022 | 0.9079±0.0013 | 0.9004±0.0012 | 0.6584±0.0054 | 0.6470±0.0051 | 0.8284±0.0034 |
| Multi-task methods | | | | | | | |
| HPS | 6.0M | 0.8052±0.0032 | 0.8740±0.0056 | 0.8759±0.0020 | 0.6093±0.0047 | 0.6097±0.0136 | 0.8109±0.0071 |
| MMoE | 37M | 0.7959±0.0094 | 0.8621±0.0051 | 0.8682±0.0103 | 0.5459±0.0065 | 0.5832±0.0123 | 0.7970±0.0047 |
| CGC | 43M | 0.7992±0.0029 | 0.8604±0.0062 | 0.8683±0.0038 | 0.5933±0.0132 | 0.6083±0.0058 | 0.8108±0.0007 |
| MTEEG-Base | 1.8M | 0.8096±0.0004 | 0.8775±0.0004 | 0.8784±0.0028 | 0.6438±0.0024 | 0.6281±0.0042 | 0.8184±0.0069 |
| MTEEG-Large | 7.4M | **0.8105**±0.0022 | **0.8801**±0.0102 | **0.8928**±0.0046 | **0.6538**±0.0066 | **0.6596**±0.0044 | **0.8321**±0.0037 |

Table 3: Results on SEED-V and CHB-MIT

| Methods | # Trainable Parameters | SEED-V | | | CHB-MIT | | |
|---|---|---|---|---|---|---|---|
| | | Balanced Acc. ↑ | Cohen's Kappa ↑ | Weighted F1 ↑ | Balanced Acc. ↑ | AUC-PR ↑ | AUROC ↑ |
| Single-task methods | | | | | | | |
| SPaRCNet | 0.79M | 0.2865±0.0022 | 0.1115±0.0034 | 0.2966±0.0031 | 0.8417±0.0036 | 0.9364±0.0022 | 0.9151±0.0039 |
| ContraWR | 1.6M | 0.3681±0.0028 | 0.2099±0.0031 | 0.3682±0.0042 | 0.8034±0.0064 | 0.9057±0.0014 | 0.8671±0.0070 |
| CNN-Transformer | 3.2M | 0.3036±0.0127 | 0.1367±0.0218 | 0.2813±0.0260 | 0.7861±0.0026 | 0.9032±0.0043 | 0.8701±0.0024 |
| FFCL | 2.4M | 0.3714±0.0047 | 0.2152±0.0084 | 0.3750±0.0087 | 0.8106±0.0072 | 0.9225±0.0063 | 0.8918±0.0095 |
| ST-Transformer | 3.5M | 0.2828±0.0025 | 0.1182±0.0036 | 0.2740±0.0045 | 0.8229±0.0027 | 0.9165±0.0047 | 0.8942±0.0058 |
| BIOT | 3.2M | 0.3831±0.0066 | 0.2238±0.0089 | 0.3831±0.0049 | 0.8439±0.0035 | 0.9367±0.0005 | 0.9026±0.0018 |
| LaBraM-Base | 5.8M | 0.4097±0.0065 | 0.2616±0.0086 | 0.4119±0.0012 | 0.8229±0.0311 | 0.9260±0.0066 | 0.8989±0.0088 |
| LaBraM-Large | 46M | 0.4188±0.0028 | 0.2733±0.0027 | 0.4253±0.0021 | 0.8653±0.0107 | 0.9346±0.0154 | 0.9166±0.0147 |
| Multi-task methods | | | | | | | |
| HPS | 6.0M | 0.4107±0.0050 | 0.2684±0.0062 | 0.4208±0.0064 | 0.7524±0.0002 | 0.9223±0.0139 | 0.8914±0.0135 |
| MMoE | 37M | 0.4113±0.0071 | 0.2651±0.0096 | 0.4182±0.0077 | 0.7221±0.0158 | 0.8994±0.0148 | 0.8572±0.0201 |
| CGC | 43M | 0.4067±0.0013 | 0.2592±0.0040 | 0.4145±0.0039 | 0.7360±0.0071 | 0.9014±0.0267 | 0.8625±0.0427 |
| MTEEG-Base | 1.8M | 0.4112±0.0028 | 0.2677±0.0037 | 0.4173±0.0035 | 0.8586±0.0152 | 0.9742±0.0015 | 0.9656±0.0025 |
| MTEEG-Large | 7.4M | **0.4226**±0.0003 | **0.2778**±0.0010 | **0.4277**±0.0018 | **0.8712**±0.0091 | **0.9779**±0.0062 | **0.9733**±0.0087 |

Table 4: Results on Sleep-EDF and PhysioNet

| Methods | # Trainable Parameters | Sleep-EDF | | | PhysioNet | | |
|---|---|---|---|---|---|---|---|
| | | Balanced Acc. ↑ | Cohen's Kappa ↑ | Weighted F1 ↑ | Balanced Acc. ↑ | Cohen's Kappa ↑ | Weighted F1 ↑ |
| Single-task methods | | | | | | | |
| SPaRCNet | 0.79M | 0.7066±0.0055 | 0.6378±0.0100 | 0.7538±0.0073 | 0.5088±0.0050 | 0.4355±0.0079 | 0.6253±0.0044 |
| ContraWR | 1.6M | 0.7148±0.0023 | 0.6785±0.0080 | 0.7837±0.0063 | 0.3855±0.0021 | 0.2673±0.0065 | 0.4888±0.0059 |
| CNN-Transformer | 3.2M | 0.7095±0.0027 | 0.6874±0.0052 | 0.7869±0.0054 | 0.3967±0.0041 | 0.2986±0.0015 | 0.5324±0.0016 |
| FFCL | 2.4M | 0.7143±0.0144 | 0.6633±0.0265 | 0.7739±0.0152 | 0.3868±0.0007 | 0.2532±0.0037 | 0.5202±0.0040 |
| ST-Transformer | 3.5M | 0.6993±0.0020 | 0.6630±0.0006 | 0.7690±0.0015 | 0.4440±0.0005 | 0.3301±0.0081 | 0.5433±0.0065 |
| BIOT | 3.2M | 0.7006±0.0014 | 0.6740±0.0096 | 0.7799±0.0065 | 0.3346±0.0006 | 0.1642±0.0061 | 0.3262±0.0313 |
| LaBraM-Base | 5.8M | 0.7003±0.0035 | 0.6742±0.0015 | 0.7789±0.0025 | 0.5072±0.0011 | 0.4303±0.0053 | 0.6110±0.0033 |
| LaBraM-Large | 46M | 0.7125±0.0050 | 0.6854±0.0006 | 0.7867±0.0034 | 0.5278±0.0017 | 0.4472±0.0089 | 0.6218±0.0059 |
| Multi-task methods | | | | | | | |
| HPS | 6.0M | 0.6628±0.0098 | 0.6411±0.0107 | 0.7647±0.0065 | 0.4571±0.0120 | 0.3677±0.0216 | 0.5679±0.0140 |
| MMoE | 37M | 0.6623±0.0113 | 0.6583±0.0128 | 0.7666±0.0070 | 0.4397±0.0059 | 0.3357±0.0017 | 0.5455±0.0019 |
| CGC | 43M | 0.6636±0.0072 | 0.6573±0.0147 | 0.7683±0.0077 | 0.5051±0.0070 | 0.4113±0.0119 | 0.5986±0.0104 |
| MTEEG-Base | 1.8M | 0.6847±0.0019 | 0.6574±0.0008 | 0.7720±0.0009 | 0.5087±0.0059 | 0.4376±0.0054 | 0.6117±0.0038 |
| MTEEG-Large | 7.4M | **0.6989**±0.0012 | **0.6645**±0.0018 | **0.7763**±0.0011 | **0.5308**±0.0055 | **0.4586**±0.0086 | **0.6315**±0.0059 |

induced by an excess of parameters. Therefore, we select $r = 8$ as the default configuration in our experiments.

**Impact of adapter locations**. The selection of locations for applying low-rank adapters is known to significantly influence the model's performance (Hu et al., 2021). Thus, we evaluate three different configurations of adapter locations: (1) only in multi-head self-attention modules (MHSA), (2) only in the feed-forward networks (FFN) that follow MHSA, (3) in both MHSA and FFN. As shown in Figure 4, the adaptations of both MHSA and FFN are crucial, as the elimination of either leads to a significant decline in performance.

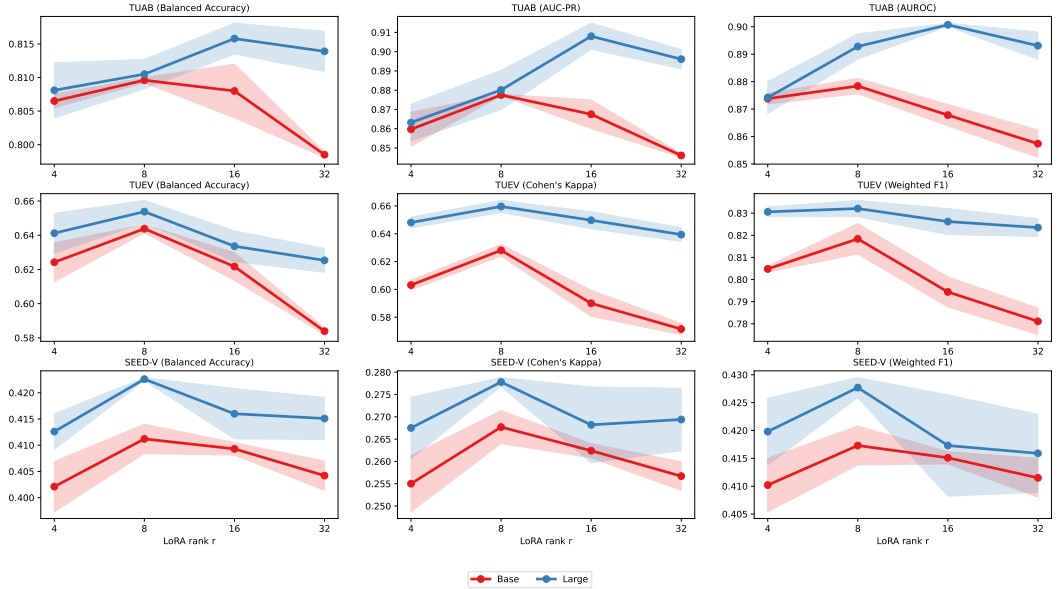

Figure 3: Ablation study on the impact of adapter rank $r$.

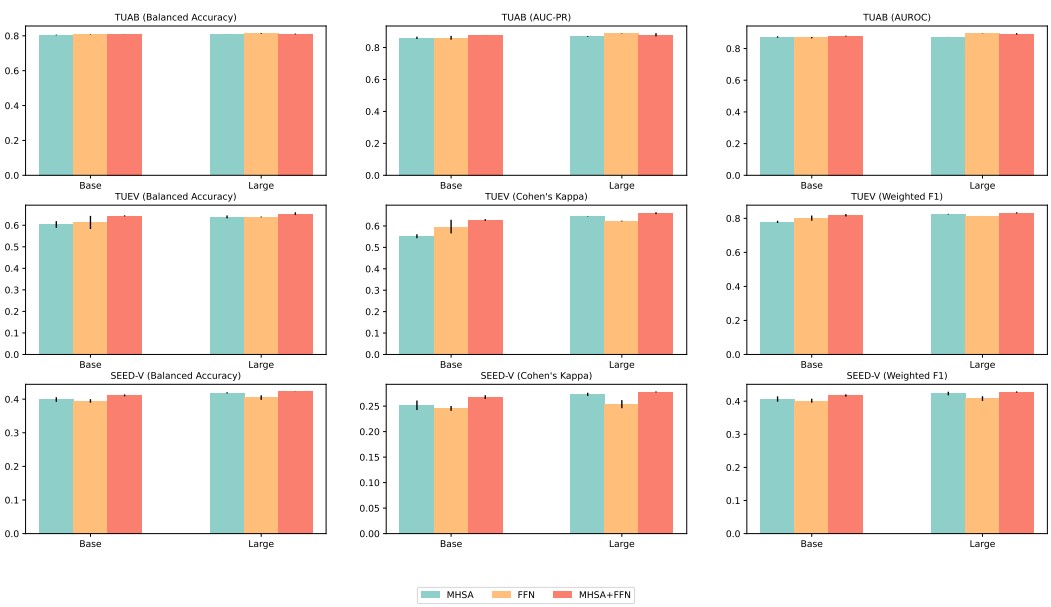

Figure 4: Ablation study on the impact of adapter locations.

**Contribution of temporal encoder**. The task-agnostic temporal encoder is designed to promote interaction among different tasks. To examine its actual contribution to the model's downstream performance, we freeze it during fine-tuning and observe the resultant impact. As shown in Figure 5, freezing the temporal encoder leads to a notable decline in performance across all the tasks and metrics, with a more pronounced decrease observed in the more challenging multi-class classifica-

tion tasks. This suggests that the temporal encoder manages to capture global knowledge that helps with reducing overfitting and enhancing the generalizability of the model.

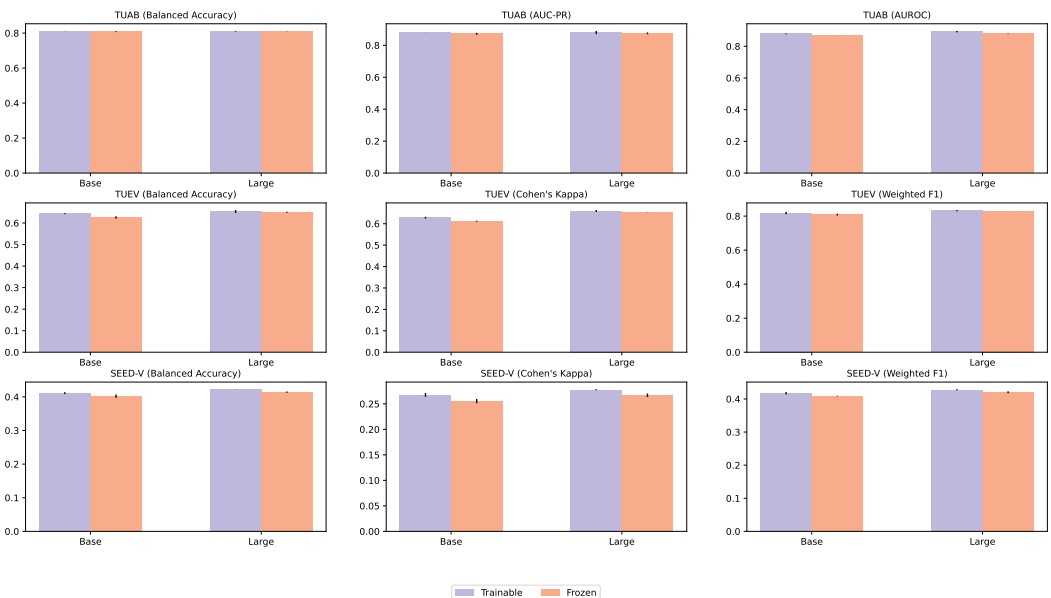

Figure 5: Ablation study on the contribution of temporal encoder.

## 5 CONCLUSION

This paper introduces MTEEG, an innovative multi-task EEG recognition framework. Utilizing a powerful pre-trained model, MTEEG incorporates a task-agnostic temporal encoder to capture global knowledge, along with task-specific low-rank adaptation modules to disentangle the parameter spaces for different tasks, thereby alleviating the conflicts stemming from the heterogeneity of EEG signals. We validate the effectiveness of MTEEG by fine-tuning it jointly on six publicly available datasets. Experiments show that MTEEG can simultaneously manage abnormal detection, event type classification, emotion recognition, seizure detection, sleep stage classification and motor imagery classification, outperforming other multi-task methods and matching the performance of state-of-the-art single-task methods. The adaptability and applicability of MTEEG demonstrate the significant potential of multi-task EEG recognition and promote the advancement of general-purpose brain-computer interfaces in the future.

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

## A  PRE-TRAINING DATASETS

We use a selection of datasets from the original LaBraM paper, omitting the private ones, for pre-training. The overall duration is approximately 2000 hours.

Table 5: Information of datasets used for pre-training.

| Dataset | #Channel | Rate (Hz) | Time (h) | Description |
|---|---|---|---|---|
| TUEP (Veloso et al., 2017) | 19-23 | 256 | 591.22 | A subset of TUEG that contains 100 subjects epilepsy and 100 subjects without epilepsy, as determined by a certified neurologist. |
| TUSL (von Weltin et al., 2017) | 23 | 256 | 20.59 | A subset of TUEG that contains annotations of slowing events. |
| TUSZ (Shah et al., 2018) | 19-23 | 256 | 1138.53 | A corpus containing EEG signals that have been manually annotated data for seizure events (start time, stop, channel and seizure type). |
| TUAR (Buckwalter et al., 2021) | 23 | 256 | 92.22 | A subset of TUEG that contains annotations of 5 different artifacts: (1) eye movement (EYEM), (2) chewing (CHEW), (3) shivering (SHIV), (4) electrode pop, electrode static, and lead artifacts (ELPP), and (5) muscle artifacts (MUSC). |
| SEED Series (Zheng & Lu, 2015; Zheng et al., 2018; Liu et al., 2022a) | 62 | 1000 | 166.75 | Emotional datasets collected when subjects watched videos. These datasets include SEED (15 subjects), SEED-IV (15 subjects), SEED-GER (8 subjects), and SEED-FRA (8 subjects). |
| Raw EEG Data (Trujillo, 2020) | 64 | 256 | 34.35 | A dataset containing EEG signals recorded during the reported Information-Integration categorization task and the reported multidimensional Rule-Based categorization task. |

## B  ADDITIONAL DETAILS OF FINE-TUNING

### B.1  DATA SPLIT

**TUAB** and **TUEV**: The training and test sets are provided by the original creator of the dataset. We adhere to BIOT and LaBraM to partition the training set into training and validation subsets at a ratio of 80% and 20%, respectively.

**SEED-V**: We divide the 15 trials of each session into three groups of five, then consolidate each group from all sessions to create the training, validation, and test sets.

**CHB-MIT**: There are a total of 23 cases collected from 22 subjects. Following BIOT, we use cases 1 to 19 for training, cases 20 and 21 for validation, and cases 22 and 23 for testing.

**Sleep-EDF** and **PhysioNet**: We partition the recordings by order into training, validation and test sets at a ratio of 64%, 16% and 20%, respectively.

## B.2 HYPERPARAMETERS

Table 6: Hyperparameters for downstream fine-tuning.

| Hyperparameters | Values |
|---|---|
| Batch size | 128 |
| LoRA learning rate | 5e-3 |
| Temporal encoder learning rate | 5e-4 |
| Minimal learning rate | 1e-6 |
| Learning rate scheduler | Cosine |
| Optimizer | AdamW |
| Adam $\beta$ | (0.9,0.999) |
| Weight decay | 0.05 |
| Total epochs | 50 |
| Warmup epochs | 5 |
| Drop path | 0.1 |
| Layer-wise learning rate decay | 0.9 |
| Label smoothing (multi-class classification) | 0.1 |

## C ADDITIONAL RESULTS OF ABLATION STUDIES

The results of ablation studies on CHB-MIT, Sleep-EDF and PhysioNet are shown in Figure 6, 7 and 8. We observe similar trends to those in Figure 3, 4 and 5, which are summarized as follows:

- MTEEG reaches peak performance when the rank of adapters is set to 8.

- Adaptations to both the MHSA and FFN modules in transformer encoder are crucial, as eliminating either of them results a significant decrease in the model's downstream performance.

- The shared temporal encoder enables interaction between different tasks, thereby reducing overfitting and further boosting the performance.

These observations are consistent across all tasks and metrics, thereby affirming their validity.

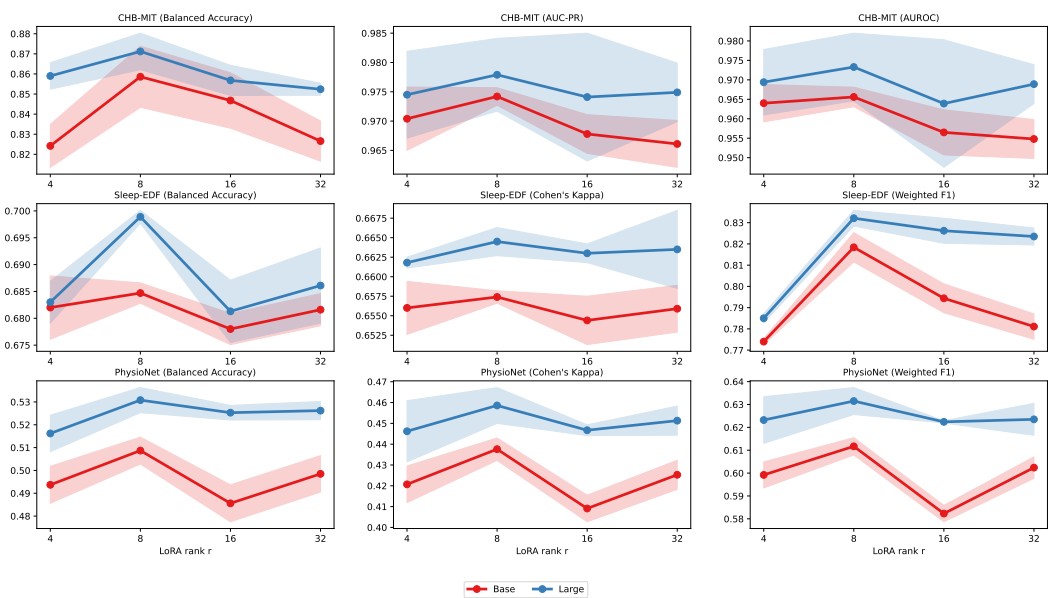

Figure 6: Additional results of ablation study on the impact of adapter rank $r$.

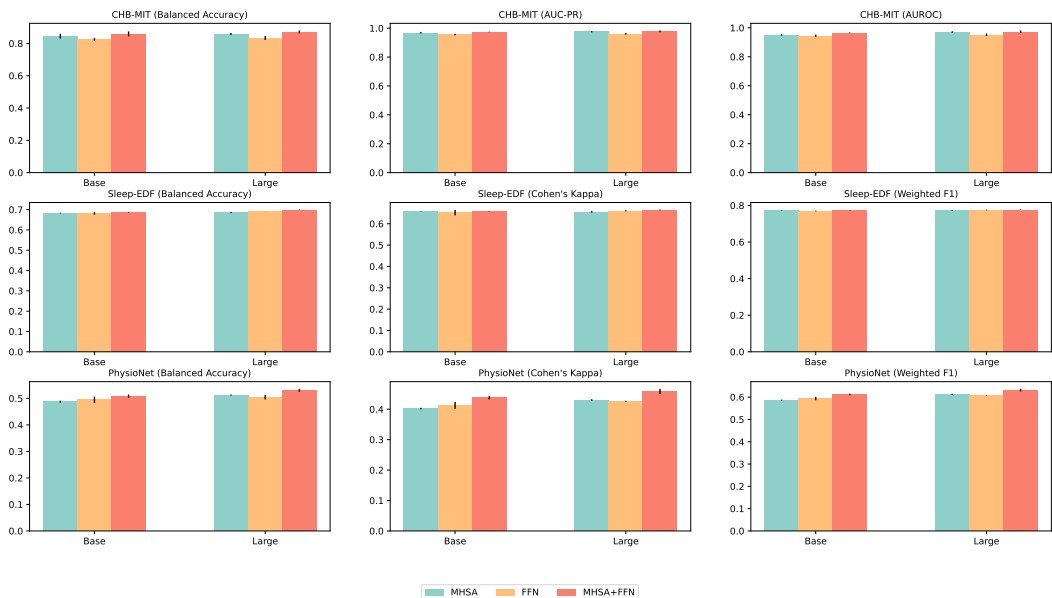

Figure 7: Additional results of ablation study on the impact of adapter locations.

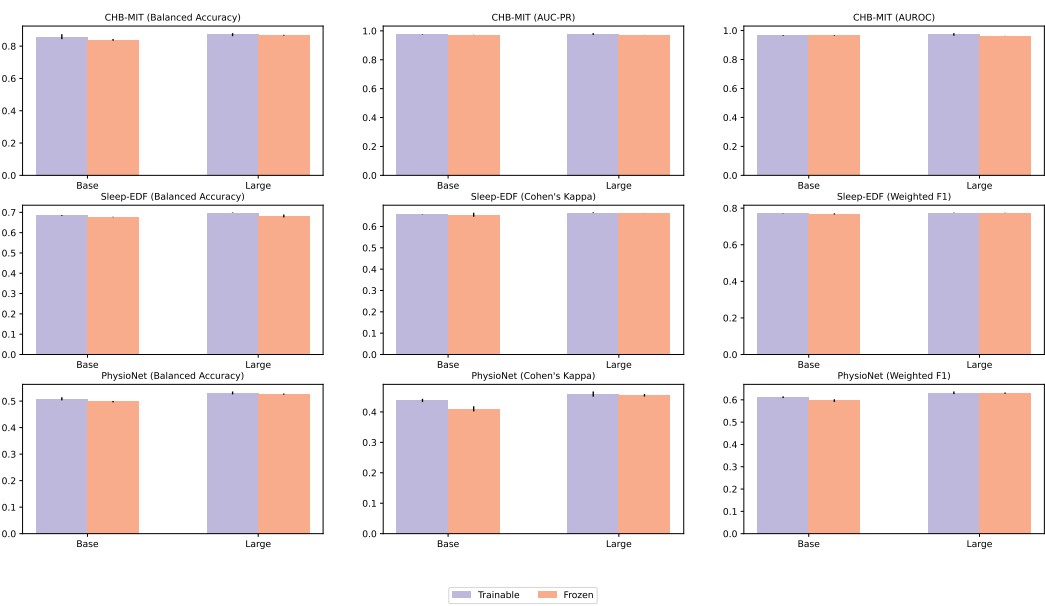

Figure 8: Additional results of ablation study on the contribution of temporal encoder.

# D DISCUSSION

MTEEG represents a groundbreaking study in the joint optimization on heterogeneous EEG datasets to facilitate multi-task capability, yielding commendable results across diverse downstream tasks. Nonetheless, we note that it has the following limitations. Firstly, the representational ability of MTEEG is significantly influenced by the selection of the pre-trained model. The pre-training phase, although not the primary focus of this paper, is an essential element that establishes the upper limit of the model's performance. Therefore, MTEEG would benefit from the future advancement of self-supervised EEG pre-training paradigms. Secondly, the EEG datasets exhibit significant variability in size and convergence speed, leading to challenges in balancing the optimization processes. In this study, we employ a rudimentary strategy to sample a subset of the data points in TUAB for each

training epoch, thereby decelerating convergence on this particular dataset; however, this approach is suboptimal and presents significant opportunities for enhancement. Looking ahead, we believe that adopting a more adaptive approach to handle the imbalance between different datasets would greatly enhance multi-task joint training.

