# OpenReview forum: "MTEEG: A Multi-Task Learning Framework for Enhanced Electroencephalography Analysis Using Low-Rank Adaptation"
_ICLR.cc/2025/Conference — Submitted to ICLR 2025_

### Official Review · Reviewer_muVQ · 2024-10-29

**Soundness:** 2
**Presentation:** 3
**Contribution:** 2
**Rating:** 6
**Confidence:** 5

**Summary:**

The paper introduces a multi-task EEG classification model by fine-tuning a pre-trained large-scale EEG model, LaBraM. The authors employ a popular fine-tuning strategy, Low-Rank Adaptation (LoRA), to adapt the pre-trained model. The model architecture consists of a temporal encoder and a transformer encoder. The temporal encoder is responsible for mapping EEG patches into embeddings, while the transformer encoder captures global features. During training, the pre-trained weights in the transformer encoder are kept frozen, and only the LoRA adapters within the transformer are trainable. The model’s effectiveness is demonstrated across six EEG datasets, each representing a different learning task.

**Strengths:**

The paper is clear and readable, with well-designed figures highlighting key points. The authors aim to explore multi-task EEG recognition by training a model capable of handling multiple tasks simultaneously. Unlike prior studies on single-task learning, this work updates and trains the model on all tasks concurrently. The experimental section is both informative and thorough, providing valuable insights. Moreover, using a large EEG dataset demonstrates the potential of leveraging large-scale EEG models in this domain.

**Weaknesses:**

The novelty of the paper is somewhat limited. Based on the experimental results, I question the necessity of multi-task learning in this context. Theoretically, due to the inherent heterogeneity of EEG data—such as differences in channel numbers, collection protocols, and medical equipment—fine-tuning all downstream tasks together might not be the most effective strategy. Additionally, your design of task-specialized LoRA for training appears nearly identical to fine-tuning each task individually. This concern is supported by the experimental results, which show that the multi-task learning approach (MTEEG) sometimes leads to only marginal improvements or even worse performance compared to single-task learning methods.

To demonstrate the necessity of multi-task fine-tuning, an ablation study should be conducted to compare fine-tuning each task individually against multi-task fine-tuning. Furthermore, it would strengthen the paper to include comparisons with TCN [1] and Medformer [2] as baseline methods. From my experience, despite their complexity, many EEG domain models struggle to outperform TCN on classification tasks. Medformer, a recent method designed specifically for EEG and ECG classification, should also be considered. Ensure that both TCN and Medformer have at least six layers and use default parameters during evaluation to maintain consistency. Due to the limited time, you can only evaluate on datasets TUEV and SEED-V.

[1] An empirical evaluation of generic convolutional and recurrent networks for sequence modeling.
[2] Medformer: A Multi-Granularity Patching Transformer for Medical Time-Series Classification.

**Questions:**

Could you provide more details about your data split? I have checked your appendix, but I'd like to know whether you are using the subject-dependent or subject-independent setup. The subject-dependent setup, while often yielding higher performance metrics, is generally inapplicable in real-world scenarios and risks potential data leakage, as the model might learn subject-specific characteristics rather than generalizable patterns. This can lead to deceptively high performance compared to the subject-independent setup.

---

> ### Author Response · Authors · 2024-11-21
> **Response to Reviewer muVQ**
>
> ## Reviewer muVQ
>
> We appreciate reviewer muVQ's insightful comments. Please see below for the point-by-point responses.
>
> > [W1] The novelty of the paper is somewhat limited. Based on the experimental results, I question the necessity of multi-task learning in this context.
>
> Following reviewer's advice, we conducted experiments on single-task fine-tuning with LoRA (with LaBraM-Base backbone), the results of which are shown in the table below. To stay consistent with the experimental settings in the main paper, we repeat the experiments with three different random seeds (0, 5, 10) and report the mean and std values.
>
> ||TUAB|TUEV|SEED-V|CHB-MIT|Sleep-EDF|PhysioNet|
> |-|-|-|-|-|-|-|
> |Balanced Acc.|0.8105 $\pm$ 0.0026|0.6431 $\pm$ 0.0003| 0.4056 $\pm$ 0.0017 | 0.8244 $\pm$ 0.0134 |0.6918 $\pm$ 0.0061|0.5022 $\pm$ 0.0042|
> |Cohen’s Kappa (or AUC-PR)|0.8944 $\pm$ 0.0059|0.6111 $\pm$ 0.0108|0.2564 $\pm$ 0.002| 0.9329 $\pm$ 0.0023|0.6689 $\pm$ 0.0048|0.4289 $\pm$ 0.0126|
> |Weighted F1 (or AUROC)|0.8854 $\pm$ 0.0071|0.8103 $\pm$ 0.0054|0.4107 $\pm$ 0.001| 0.9105 $\pm$ 0.0021|0.7759 $\pm$ 0.0026|0.6086 $\pm$ 0.0074|
>
> From the table, we can see that the performance of single-task fine-tuning with LoRA is pretty similar to that of single-task full fine-tuning (the LaBraM-Base rows in Table 2-4), and multi-task fine-tuning is generally comparable to single-task fine-tuning, i.e, better on four datasets and worse on the other two, which, in the reviewer's opinion, indicates that multi-task fine-tuning lacks necessity. However, we would like to advocate multi-task fine-tuning for the following reasons.
> - **The performance of multi-task fine-tuning is actually good.** If we look at the research works on multi-task learning in other domains, such as image or language processing, it's very common that a newly proposed method is better than the baselines on some tasks but worse on some others, so researchers often use the average performance across all the tasks as an "ultimate metric" to demonstrate their methods' superiority. We didn't do this because we follow previous EEG works to use different performance metrics for different tasks (datasets), so it doesn't make much sense to take the average of values with different meanings. That being said, balanced accuracy is a common metric that we use for all the tasks, so we report the average values of it under three different settings in the table below. As we can see, the performance of multi-task fine-tuning is consistently better than the single-task counterparts.
>
> ||Single-task full fine-tuning|Single-task LoRA|MTEEG (multi-task LoRA)|
> |-|-|-|-|
> |Balanced Acc.|0.6441|0.6460|0.6528|
> - **Multi-task fine-tuning is beneficial for real-world deployment.** Setting the performance aside, multi-task learning is still meaningful because of its efficiency. It saves space, computation, time, and in real-life applications, it's convenient to have one unified system that can handle all the tasks simultaneously. As an example, researchers in the computer vision community used to develop separate models for classification, detection, and segmentation tasks but nowadays most of them have turned their attention to multi-task models, even though the performance of a multi-task model is not always better than a task-specific model. Currently, EEG recognition is still dominated by task-specific models, so we would like to make the first try. Hopefully, more researchers will pay attention to multi-task models in the future and develop models that are both powerful and efficient enough to be deployed in real-life BCI devices.
>
> In addition, we also conducted experiments with TCN and Medformer on TUEV and SEED-V, as suggested by the reviewer. As shown in the table below, TCN performs really well on TUEV, but both models are not good at emotion recognition on SEED-V.
>
> ||TUEV|SEED-V|
> |-|-|-|
> |TCN|0.6364 $\pm$ 0.0054, 0.6720 $\pm$ 0.0064, 0.8110 $\pm$ 0.0052|0.3200 $\pm$ 0.0048, 0.1665 $\pm$ 0.0073, 0.3131 $\pm$ 0.0061|
> |Medformer|0.4552 $\pm$ 0.0083, 0.4113 $\pm$ 0.0079, 0.6728 $\pm$ 0.0076|0.3019 $\pm$ 0.0047, 0.1291 $\pm$ 0.0083, 0.2818 $\pm$ 0.0064|
>
> > [Q1] Could you provide more details about your data split? I'd like to know whether you are using the subject-dependent or subject-independent setup
>
> We totally agree with the reviewer on that a subject-dependent setup focuses too much on the performance numbers rather than applicability in real-world scenarios, so in this work, we use a subject-independent setup for all the datasets except for SEED-V. We chose to use a subject-dependent setup for SEED-V simply because we wanted to stay consistent with previous works, especially LaBraM, which is the basis of this work and a major baseline that we compare with.

---

> > ### Comment · Reviewer_muVQ · 2024-11-21
> >
> > Thanks for the update. All my concerns have been addressed. I raised my score.

---

### Official Review · Reviewer_2WaD · 2024-10-31

**Soundness:** 3
**Presentation:** 3
**Contribution:** 1
**Rating:** 5
**Confidence:** 4

**Summary:**

This paper introduces MTEEG, a framework for multi-task EEG recognition that leverages low-rank adaptation (LoRA) to efficiently fine-tune a pre-trained EEG foundation model across multiple tasks. The study presents a pioneering attempt to apply parameter-efficient fine-tuning techniques, specifically LoRA, to the domain of EEG-based multi-task learning. MTEEG demonstrates its capability by outperforming established multi-task and single-task models across six different EEG datasets, covering tasks like abnormal detection, event classification, emotion recognition, and seizure detection.

**Strengths:**

1. This paper makes a commendable and pioneering effort in exploring parameter-efficient fine-tuning techniques, specifically Low-Rank Adaptation (LoRA), for EEG foundation models in multi-task learning (MTL) scenarios.
2. The experiments are robust, with adequate coverage of tasks and inclusion of critical ablation studies that help clarify the contribution of LoRA and other model components.

**Weaknesses:**

1. While the integration of LoRA into EEG models is an interesting step, the contribution appears limited. The paper does not introduce significant adaptations or improvements over the original LoRA method beyond its direct application in EEG-based MTL. This straightforward application of LoRA does not meet the high technical bar expected at ICLR.
2. The paper benchmarks the proposed MTEEG framework against older multi-task learning methods like HPS, MMoE, and CGC, which, while relevant, are not cutting-edge.
3. While LoRA is effectively evaluated, the paper does not explore how other advanced parameter-efficient fine-tuning methods could perform in the same context. For a more thorough assessment, it would be valuable to demonstrate if techniques like Adapters, Prefix-tuning, or Prompt Tuning could seamlessly replace LoRA or be integrated into the MTEEG framework. A more comprehensive evaluation, incorporating these baselines, would provide stronger evidence of the proposed framework's performance.

**Questions:**

This paper uses LoRA for multi-task fine-tuning across multiple datasets. However, have you considered a simpler approac, that is using a task-specific classification head for each dataset while fine-tuning the entire pre-trained LaBraM model? This method would be a straightforward way to enable multi-task learning, and I'm curious how its results compare to the LoRA-based approach.

---

> ### Author Response · Authors · 2024-11-21
> **Response to Reviewer 2WaD**
>
> We thank reviewer 2WaD for the thorough review and valuable suggestions. Please see below for the point-by-point responses.
>
> > [W1] The contribution appears limited
>
> We acknowledge that the core techniques employed in MTEEG, such as pre-trained models and LoRA, are not novel, but still would like to highlight the significant contributions of our work as follows:
> - Despite its importance, multi-task learning (MTL) has remained unexplored in the field of EEG recognition. Although some previous works [1][2][3] claim to have incorporated MTL into their frameworks, the term “task” in their works refer to the hand-crafted pretext tasks that are used to improve optimization on a single dataset, while in our work, the term “task” refers to the actual downstream tasks. Therefore, MTEEG is the first framework which enables multi-task capability by training on a wide range of different datasets simultaneously.
> - The lightweight design of MTEEG ensures both space and time efficiency, which will greatly benefit the deployment of pre-trained EEG models in real-world scenarios. Moreover, MTEEG achieves this efficiency without sacrificing performance, as proven by the fact that it performs comparably with LaBraM (fine-tuned in single-task settings).
>
> > [W2] The paper benchmarks the proposed MTEEG framework against older multi-task learning methods like HPS, MMoE, and CGC, which, while relevant, are not cutting-edge.
>
> We chose these as the multi-task baselines in our work for the following reasons.
> - Since MTEEG is a pioneering work in exploring multi-task learning for EEG recognition, there’s currently no public method in this domain to compare with, so we decided to integrate LaBraM into some general MTL frameworks to set up the multi-task baselines. Although some more recently proposed methods have demonstrated good performance on image datasets, they often incorporate architecture-specific designs, making it unclear how they could be effectively consolidated with LaBraM for applications in the EEG domain. In contrast, while methods like HPS, MMoE and CGC are old, they can be seamlessly employed to any backbone networks, including LaBraM, without any architecture-specific designs.
> - The performance of these methods are still good enough that many recent papers [4][5] include them as the baselines.
> - These methods have been re-implemented by a group of researchers and integrated into the LibMTL library. By using this library, we can ensure fair comparison and reproducibility.
>
> > [W3] For a more thorough assessment, it would be valuable to demonstrate if techniques like Adapters, Prefix-tuning, or Prompt Tuning could seamlessly replace LoRA or be integrated into the MTEEG framework.
>
> We greatly appreciate this valuable suggestion and will add these baselines to strengthen our work.
>
> > [Q1] Have you considered a simpler approach, that is using a task-specific classification head for each dataset while fine-tuning the entire pre-trained LaBraM model?
>
> We thank the reviewer for pointing this out. Yes, we have considered this simpler approach, and this is exactly what HPS (Hard Parameter Sharing) does. In Section 4.3 of the paper, we attribute the performance gap between HPS and single-task LaBraM-Base to the conflicts between different datasets, since the architectures of these approaches are almost identical, except that HPS has more classification heads.
>
> References
>
> [1] Autthasan, Phairot, et al. "MIN2Net: End-to-end multi-task learning for subject-independent motor imagery EEG classification." IEEE Transactions on Biomedical Engineering 69.6 (2021): 2105-2118.
>
> [2] Ditthapron, Apiwat, et al. "Universal joint feature extraction for P300 EEG classification using multi-task autoencoder." IEEE access 7 (2019): 68415-68428.
>
> [3] Li, Yang, et al. "GMSS: Graph-based multi-task self-supervised learning for EEG emotion recognition." IEEE Transactions on Affective Computing 14.3 (2022): 2512-2525.
>
> [4] Chang, Jianxin, et al. "Pepnet: Parameter and embedding personalized network for infusing with personalized prior information." Proceedings of the 29th ACM SIGKDD Conference on Knowledge Discovery and Data Mining. 2023.
>
> [5] Cao, Jiangxia, et al. "Towards universal cross-domain recommendation." Proceedings of the Sixteenth ACM International Conference on Web Search and Data Mining. 2023.

---

> > ### Comment · Reviewer_2WaD · 2024-11-26
> >
> > I appreciate the authors' detailed rebuttal. W2 and Q1 have been addressed. While I acknowledge that this is a pioneering effort in the EEG field to explore parameter-efficient tuning using large pre-trained models, I still find the technical contribution to be limited, as it largely revolves around a straightforward adaptation of LoRA. Overall, this paper seems borderline, with a rating likely between 5 and 6.

---

### Official Review · Reviewer_zWj6 · 2024-11-03

**Soundness:** 2
**Presentation:** 2
**Contribution:** 2
**Rating:** 5
**Confidence:** 5

**Summary:**

The article introduces MTEEG, a multi-task EEG recognition system that improves the adaptation of a pre-trained model to different tasks. MTEEG uses a pre-trained model in combination with a task-independent temporal encoder to capture global knowledge from EEG signals, while task-specific low-rank adaptation modules are employed to manage the different parameter spaces for each task. The training process of MTEEG takes place in two stages: In the first stage, a LaBraM model is pre-trained on unlabeled data to extract information from the raw EEG signals. The model is then fine-tuned for specific downstream datasets, while specific low-rank adapters for each task are integrated into the transformer encoder to ensure parameter isolation. The proposed approach was verified on 6 different datasets.

**Strengths:**

Although the authors used different data sets and conducted extensive experiments and ablation studies, the approach used is not particularly novel. A more innovative method or perspective would enhance the contribution of the research to the existing literature.

**Weaknesses:**

The paper lacks clarity and self-containment, making it difficult for readers to fully grasp the research objectives and methodologies.

The paper appears to be a combination of existing methodologies, such as employing large pre-trained models and fine-tuning layers for specific tasks. However, it lacks a significant novel contribution to the field, which limits its impact and originality in advancing the research.

The authors should clarify how their work differs from transfer learning, which typically involves freezing certain layers of a pre-trained model while training the remaining layers for a specific task. A comparison highlighting the unique aspects of their approach compared to standard transfer learning methods would improve understanding of their contribution to the field.

**Questions:**

The authors should clarify how their work differs from transfer learning, which typically involves freezing certain layers of a pre-trained model while training the remaining layers for a specific task.

If the different data sets show variations in the number of channels and sampling frequencies of the EEG signals, the authors should provide a clear explanation of how they resampled the EEG signals to 200 Hz. This should include details of the methods used for upsampling and downsampling to ensure consistency between data sets.

---

> ### Author Response · Authors · 2024-11-21
> **Response to Reviewer zWj6 (Part 1/2)**
>
> We are grateful for reviewer zWj6’s careful review and insightful comments. Please see below for the point-by-point responses.
>
> > [W1] The paper lacks clarity and self-containment, making it difficult for readers to fully grasp the research objectives and methodologies.
>
> We thank the reviewer for pointing this out, which will help us improve the clarity of our paper. The fundamental idea behind this work is that while current EEG pre-training models are powerful, they require full fine-tuning to perform well on a downstream task. In a clinical environment, we may need many types of information (e.g. seizure, sleep, emotion, etc.) to be simultaneously decoded from the patient’s brain signal to comprehensively evaluate the patient’s health condition. In this case, a straightforward solution would be to replicate and fine-tune a pre-trained model multiple times, once for each type of information (task). However, doing this will introduce significant space (to store all the replicas) and computation/time (to optimize all the replicas) overhead. Therefore, the primary question we want to ask here is: Can we just have one unified model that can handle all the tasks simultaneously? To achieve this, the model needs to be fine-tuned simultaneously on multiple different datasets, which poses challenges since the heterogeneity can cause conflicts in the optimization directions. A direct consequence of the conflict issue is a significant drop in downstream performance, which is shown by the performance gap between HPS and LaBraM-Base in Table 2, 3, and 4. HPS (hard parameter sharing) here refers to the naïve implementation where we attach task-specific classification heads to a pre-trained LaBraM-Base and directly fine-tune the whole model on all datasets. To mitigate the conflict issue, we apply the task-specific LoRAs to isolate the parameter space, while keeping some modules task-independent so that different tasks can still benefit from some common knowledge. Overall, 1) The proposed MTEEG is more space and computation/time efficient, which addresses the problem of fine-tuning a pre-trained model multiple times as described above. 2) MTEEG achieves this efficiency without sacrificing performance, as proven by the fact that it performs comparably with LaBraM (fine-tuned in single-task settings).
>
> > [W2] The paper appears to be a combination of existing methodologies and lacks a significant novel contribution to the field
>
> We acknowledge that the core techniques employed in MTEEG, such as pre-trained models and LoRA, are not novel, but still would like to highlight the significant contributions of our work as follows:
> - Despite its importance, multi-task learning (MTL) has remained unexplored in the field of EEG recognition. Although some previous works [1][2][3] claim to have incorporated MTL into their frameworks, the term “task” in their works refer to the hand-crafted pretext tasks that are used to improve optimization on a single dataset, while in our work, the term “task” refers to the actual downstream tasks. Therefore, MTEEG is the first framework which enables multi-task capability by training on a wide range of different datasets simultaneously.
> - As mentioned in our response to the previous question, the lightweight design of MTEEG ensures both space and time efficiency, which will greatly benefit the deployment of pre-trained EEG models in real-world scenarios.
>
> > [W3] The authors should clarify how their work differs from transfer learning
>
> We thank the reviewer for pointing out the need for this clarification. Transfer learning is a general concept which, to our knowledge, refers to the process of adapting a pre-trained model to a given target task. In this sense, our work is a special form of transfer learning. There are two major differences between MTEEG and a standard/typical implementation of transfer learning.
> - Transfer learning typically involves only one downstream dataset at a time, but MTEEG fine-tunes a pre-trained model on multiple datasets concurrently. This poses a unique challenge regarding the conflicts between different datasets, so the model needs to be carefully designed to reach a balance between what should be shared versus what should be disentangled.
> - As the reviewer mentioned, a common practice in transfer learning is to freeze some layers and train the others for a specific task. However, current EEG pre-training models require full fine-tuning to achieve good downstream performance. Thus, MTEEG does not “freeze” any layer conceptually, it just utilizes the LoRA modules to approximate the changes of all the parameters.

---

> ### Author Response · Authors · 2024-11-21
> **Response to Reviewer zWj6 (Part 2/2)**
>
> > [Q1] The authors should clarify how their work differs from transfer learning
>
> This question turns out to be a duplicate of W3. Please refer to our response above.
>
> > [Q2] The authors should provide a clear explanation of how they resampled the EEG signals to 200 Hz
>
> We agree with the reviewer on that the resampling strategy should be consistent across datasets. In this work, the MNE-Python [4] library is used for all the preprocessing work, including resampling. We called the resample function with default parameters, that is, `resample(sfreq, *, npad='auto', window='auto', stim_picks=None, n_jobs=None, events=None, pad='auto', method='fft', verbose=None)`. Looking into the source code of the library, we found that the resample function first pads the input so that the sequence length is a power of two, then creates a boxcar window, and calls `scipy.signal.resample()` function to perform resampling with the Fourier-based method.
>
> References
>
> [1] Autthasan, Phairot, et al. "MIN2Net: End-to-end multi-task learning for subject-independent motor imagery EEG classification." IEEE Transactions on Biomedical Engineering 69.6 (2021): 2105-2118.
>
> [2] Ditthapron, Apiwat, et al. "Universal joint feature extraction for P300 EEG classification using multi-task autoencoder." IEEE access 7 (2019): 68415-68428.
>
> [3] Li, Yang, et al. "GMSS: Graph-based multi-task self-supervised learning for EEG emotion recognition." IEEE Transactions on Affective Computing 14.3 (2022): 2512-2525.
>
> [4] Gramfort, Alexandre, et al. "MEG and EEG data analysis with MNE-Python." Frontiers in Neuroinformatics 7 (2013): 267.

---

### Official Review · Reviewer_oDYw · 2024-11-04

**Soundness:** 2
**Presentation:** 2
**Contribution:** 2
**Rating:** 3
**Confidence:** 4

**Summary:**

MTEEG is proposed to solve the problem of fine-tuning a pretrained feature extractor separately for multiple downstream tasks, with the goal of reducing the computational cost of fine-tuning multiple times. MTEEG uses task-agnostic EEG encoders (temporal) and task-specific LoRA adaptors on top of a versatile pre-trained EEG backbone (LaBraM). Results on unseen tasks/datasets indicate MTEEG either maintains or surpasses single-task fine-tuning performance while using fewer overall parameters.

**Strengths:**

- MTEEG highlights the possibility of shared task structure/knowledge in heterogeneous EEG task/application domains that can be exploited during modeling, especially in the emerging EEG + AI space.
- Use of LoRA leads to parameter efficiency compared to base EEG "foundation" models, which is a desirable property for deployment.

**Weaknesses:**

- Preprocessing of the EEG signal needs improvement: 1) physiologic signal content of scalp EEG is between 0.1-25Hz (lower frequencies). Anything beyond 25Hz (45Hz at most) is considered high-frequency noise. 2) notch filter depends on which country the data was collected in. Example -- TUAB/TUEV are collected in US where power line freq. is 60Hz. 3) Sampling frequency can be safely dropped to 128Hz or even 80Hz, which will save computation time. 3) EEGs ideally would be normalized channel-wise (using statistics calculated from the whole recording), otherwise amplitude variability across channels is lost. Consider approach in [1] for preprocessing.
- Reliability of results: 1) Variability that is important to report is of the test set (perhaps bootstrap samples?) rather than random seeds (Tables 2-4). 2) Significance testing of mean/group differences is needed to trust the ablation results (Figures 4, 5).
- As far as I understand, the proposed approach is only applicable when all downstream tasks of interest are known beforehand. As such, the problem/motivation/hypothesis and real-world need/utility are unclear to me (line 15-20, line 48-53). In future clinical environments, we may want to have multiple EEG models/tasks "active" at the same time rather than switch from one model/task to another, while the EEG spatial configuration remains the same (10-20 system), i.e., there is no spatial heterogeneity. Secondly, fine-tuning is done using few task labels given that EEG labels are very expensive to obtain. Therefore, convergence is typically achieved relatively quickly, i.e., the time/computation overhead is practically negligible to fine-tune a backbone separately for each task of interest.
- Overall presentation can be substantially improved: 1) discussion and limitations should be substantial, insightful, and be in the main text, 2) figures are taking up too much space for the insight they provide (can start the y-axis at 0.5 instead of 0.0), 3) the core argument/narrative of need of multi-task learning in EEG and central hypothesis/question of the study needs revision and clarity.


[1] Banville, Hubert, et al. "Uncovering the structure of clinical EEG signals with self-supervised learning." Journal of Neural Engineering 18.4 (2021): 046020.

**Questions:**

- Its unclear to me what the central hypothesis and question of the study is. Is it simply about saving time during fine-tuning or inference when all downstream tasks are known apriori?
- Typo in equation at line 170: Do you mean C_p instead of C_k?
- Q: What exactly happens in line 174? Do you mean temporal and spatial positional encodings?
- p refers to tasks (line 208, figure 2) and datasets (line 154) interchangeably, which is quite confusing. You can explain/justify the equality at the beginning and use either meaning consistently throughout the paper.
- Q: is the variability shown in Figures 4 and 5 statistically significant?
- Q: Is the parameter efficiency of MTEEG only due to the low-rank adaptor module, or are there other differences b/w LaBraM-base and MTEEG-base?
- Consider adding experiments that can show the heterogeneous conflict issue in multi-task fine-tuning (line 59-60) without LoRA adaptors and how the knowledge isolation is acheived using LoRA in MTEEG? I think this clarity is needed for readers to fully understand and "see" the issue the paper claims to address.
- These studies may be of interest for future work: 1) for self-supervised EEG pre-training and generalizability [1][2] and 2) effect of EEG heterogeneity/variability on EEG-ML/AI models [3].


[1] Banville, Hubert, et al. "Uncovering the structure of clinical EEG signals with self-supervised learning." Journal of Neural Engineering 18.4 (2021): 046020.

[2] Wagh, Neeraj, et al. "Domain-guided self-supervision of eeg data improves downstream classification performance and generalizability." Machine Learning for Health. PMLR, 2021.

[3] Wagh, Neeraj, et al. "Evaluating latent space robustness and uncertainty of EEG-ML models under realistic distribution shifts." Advances in Neural Information Processing Systems 35 (2022): 21142-21156.

---

> ### Author Response · Authors · 2024-11-21
> **Response to Reviewer oDYw (Part 1/2)**
>
> We thank reviewer oDYw for providing a thorough review with constructive comments. Please see below for the point-by-point responses.
>
> > [W1] Preprocessing of the EEG signal needs improvement.
>
> We thank the reviewer for providing the valuable suggestions. The reason why we chose the current preprocessing strategy is two-fold. Firstly, the same strategy has been applied in LaBraM[1], which is the framework that we build our method upon and compare our method to. We would like to stay consistent with LaBraM so that it makes more sense to attribute the performance difference to our core contribution (multi-task design) instead of other aspects such as preprocessing and data split. Secondly, since we are dealing with large-scale heterogeneous data, we are inclined to be conservative by only applying minimal preprocessing. While we agree with the reviewer on that anything beyond 25Hz (45Hz at most) is usually considered high-frequency noise, there are studies suggesting that higher frequencies could be useful [2]. It’s hard to tell what the optimal preprocessing strategy is for each of the datasets, so we decided to not filter out that much information and let the model itself determine what is important. Regarding the notch filter, we acknowledge it’s our oversight to carry this over. While the original authors of LaBraM think this is not that big of an issue [3], we thank the reviewer for pointing it out and helping us improve the quality of our work.
>
> > [W2] Reliability of results
>
> We wonder if the reviewer could elaborate more on what type of variability is more important to report in the result section. To our knowledge, it seems common [1][4][5] to report the variability of performance metrics based on different random initializations of the model (with different random seeds) to ensure that the reported performance gain or loss is not simply due to randomness.
>
> We thank the reviewer for highlighting the need for significance testing to support the ablation results. We performed t-test on the results in Figure 4 and 5 and show the p-values below. The three values in each cell correspond to the three performance metrics respectively.
>
> ||TUAB|TUEV|SEED-V|
> |-|-|-|-|
> |MHSA v.s. MHSA+FFN (base)|0.0357 0.2655 0.2301|0.0224  0.0008  0.0096|0.0779  0.0871  0.1597|
> |FFN v.s. MHSA+FFN (base)|0.1154 0.4898 0.0571|0.2300  0.4085  0.4888|0.0176  0.0067  0.0327|
> |MHSA v.s. MHSA+FFN (large)|0.1721  0.9576  0.5987|0.0730  0.0095  0.0581|0.0216  0.0759  0.2408|
> |FFN v.s. MHSA+FFN (large)|0.2278  0.3016  0.3940|0.0361  0.0004  0.0031|0.0102  0.0065  0.0100|
>
> ||TUAB|TUEV|SEED-V|
> |-|-|-|-|
> |Trainable v.s. Frozen (base)|0.4622  0.0177  0.888|0.0074  0.1294  0.7981|0.0562  0.0258  0.0342|
> |Trainable v.s. Frozen (large)|0.688  0.8896  0.3154|0.9898  0.0884  0.4797|0.0026  0.0074  0.0248|
>
>
> > [W3] The problem/motivation/hypothesis and real-world need/utility are unclear to me
>
> We greatly appreciate the reviewer for helping to clarify what the actual need is in future clinical environments - having multiple EEG models/tasks "active" at the same time rather than switch from one model/task to another, so we have adjusted the manuscript accordingly to make this clear and hopefully avoid confusion in the future.
>
> Regarding the second point, i.e., the time/computation overhead of fine-tuning a separate backbone for each task of interest. While this overhead can be negligible if the dataset and the model are both small enough, we argue that this will not be the case in the future, especially with the rapid advancements of neural signal collection and AI techniques. As a reference, the TUAB dataset used in this current study contains 409,455 10-second samples, and it takes 3:20:27 to fine-tune a LaBram-Base (5.8M) on it for 50 epochs with a batch size of 256 using two NVIDIA A100 GPUs. People may have different perspectives, but we already would not consider this time/space/computation cost as negligible, not to say that works in this field have recently scaled EEG pre-training models up to have billions of parameters [6].
>
> > [W4] Overall presentation can be substantially improved
>
> We thank the reviewer for these valuable tips and will make changes accordingly in the revision.

---

> ### Author Response · Authors · 2024-11-21
> **Response to Reviewer oDYw (Part 2/2)**
>
> > [Q1] Its unclear to me what the central hypothesis and question of the study is. Is it simply about saving time during fine-tuning or inference when all downstream tasks are known apriori?
>
> Yes, time, space, and computation efficiency are valuable features that will greatly facilitate real-world deployment. Moreover, MTEEG achieves this efficiency without sacrificing performance.
>
> > [Q2] Typo in equation at line 170: Do you mean C_p instead of C_k?
>
> Yes, it should be $C_p$ instead of $C_k$. We thank the reviewer for pointing out this typo and will correct it in the revision.
>
> > [Q3] What exactly happens in line 174? Do you mean temporal and spatial positional encodings?
>
> Yes, temporal and spatial embeddings in line 174 refer to the positional encodings. We omitted some details here to save space since the positional encodings are carried over from the original LaBraM paper. Sorry for the confusion.
>
> > [Q4] p refers to tasks (line 208, figure 2) and datasets (line 154) interchangeably, which is quite confusing. You can explain/justify the equality at the beginning and use either meaning consistently throughout the paper.
>
> We are grateful for the reviewer's valuable suggestion and will add explanation for this in the revision.
>
> > [Q5] Is the variability shown in Figures 4 and 5 statistically significant?
>
> This question seems to be a duplicate of [W2]. Please refer to our response above.
>
> > [Q6] Is the parameter efficiency of MTEEG only due to the low-rank adaptor module, or are there other differences b/w LaBraM-base and MTEEG-base?
>
> Yes, the only structural difference between these two is the LoRA modules.
>
> > [Q7] Consider adding experiments that can show the heterogeneous conflict issue in multi-task fine-tuning
>
> We appreciate this really nice suggestion and will definitely consider it.
>
>
> **References**
>
> [1] Jiang, Wei-Bang, Li-Ming Zhao, and Bao-Liang Lu. "Large brain model for learning generic representations with tremendous EEG data in BCI." arXiv preprint arXiv:2405.18765 (2024).
>
> [2] Schirrmeister, Robin Tibor, et al. "Deep learning with convolutional neural networks for EEG decoding and visualization." Human brain mapping 38.11 (2017): 5391-5420.
>
> [3] https://openreview.net/forum?id=QzTpTRVtrP
>
> [4] Yang, Chaoqi, M. Westover, and Jimeng Sun. "Biot: Biosignal transformer for cross-data learning in the wild." Advances in Neural Information Processing Systems 36 (2024).
>
> [5] Cui, Wenhui, et al. "Neuro-GPT: Towards A Foundation Model For EEG." 2024 IEEE International Symposium on Biomedical Imaging (ISBI). IEEE, 2024.
>
> [6] Yuan, Zhizhang, et al. "Brant-2: Foundation Model for Brain Signals." arXiv preprint arXiv:2402.10251 (2024).

---

### Meta-Review · Area_Chair_ZgY3 · 2024-12-16

**Metareview:**

This paper presents a method for fine-tuning a pre-trained EEG model  for priori known multiple downstream tasks, instead of fine-tuning the model on each downstream dataset independently. The proposed model uses task-agnostic temporal encoders and task-specific LoRA adaptation modules on the top of a pre-trained EEG backbone.  The paper is well written and demonstrate a usefulness of multi-task fine-tuning for a pre-trained EEG model.  However, there are a few concerns that should be considered for future submissions. First of all, the benefit or necessity of multi-task fine-tuning is questionable, due to inherent heterogeneity of EEG data. Thus, fine-tuning multiple heterogeneous downstream tasks together might not be the most effective way in EEG domain. Most of reviewers also have concerns in the limited technical novelty as well as ablation study and comparison with single-task training in experiments. Therefore, the paper is not recommended for acceptance in its current form. I hope authors found the review comments informative and can improve their paper by addressing these carefully in future submissions.

**Additional Comments On Reviewer Discussion:**

During the rebuttal period, the authors made efforts in responding to issues raised by reviewers. Some of them was successful and one reviewer raised his/her score. However, most of reviewers stood by their original decision, claiming that the current paper does not meet the standard of ICLR.

---

### Decision · Program_Chairs · 2025-01-22

Reject